# Research on the Load Distribution Strategy to Meet the QoE Requirements for Conversational Real-Time HD Video Service

**Yuzhuo Zhan [1],\*** **, Weimin Lei [1],\*, Yunchong Guan [2]** and **Hao Li [1]**

1   School of Computer Science and Engineering, Northeastern University, Shenyang 110819, China; lihaozhengze@hotmail.com

2   School of Computer Science, Shenyang Aerospace University, Shenyang 110136, China; y.c.guan@foxmail.com

\*   Correspondence: 1210422@stu.neu.edu.cn (Y.Z.); leiweimin@ise.neu.edu.cn (W.L.)

**Abstract:** A reliable transmission with QoE (Quality of Experience) guarantee is crucial for internet conversational service applications. However, due to the limited network bandwidth bottleneck effect and the drawback of transmission technology, there exists no mature and open QoE technical solution for this service. In this paper, we focused our attention on a load distribution strategy for multipath relay transmission to meet the QoE requirements of conversational real-time HD video services. It consisted of three stages: First, a series of relay nodes was deployed in the backbone network, and a software defined overlay network was constructed to perform the multipath relay transmission for the service. Second, by an analysis of the QoE feature, a bijection was built for each quantitative QoE and its MOS (Mean Opinion Score) score. Finally, considering the influence of the coupling relation between paths on the service quality in multipath relay transmission, fuzzy cooperative game theory was used to design the service load distribution strategy. Many experiments showed that compared with state-of-the-art methods in the single-path transmission scene, the strategy we designed can dynamically adjust the load distribution of each sub-path according to the change in QoS (Quality of Service) information of the transmission path in real time. While meeting the strict real-time constraints of the service, it can effectively avoid the impact of network random congestion on the service QoE.

**Keywords:** conversational real-time HD video; overlay network; multipath relay service; QoE evaluation; QoS indexes; load distribution strategy; fuzzy cooperative game theory

## 1. Introduction

Quality of experience (QoE), as an evaluation method to measure the quality of a network service, is a description of the satisfaction degree of users' network service experience [1]. At present, the emergence of application-layer optimized transmission technologies, such as P2P (peer-to-peer) and CDNs (content delivery networks), has already realized transmission services that meet the QoE needs of users for a large number of web applications and streaming media services. However, with the rise of conversational real-time HD video service, such as internet phone or video conferencing, due to the fact of its strict real-time constraints, the bottleneck effect of network bandwidth, transmission technology, and other factors, there is no mature or widely used technical scheme for researching the QoE of a service.

For a conversational real-time HD video service, in the traditional single-path end-to-end transmission process, the internet's "best effort" transmission mode cannot avoid the impact of random network congestion on the quality of service, but multipath relay transmission technology can realize

the application layer media relay service without changing the existing network architecture. In theory, as long as the access network of the service terminal is not the bottleneck of transmission, when the default path causes random congestion or link interruption due to the lack of bandwidth, the multipath relay transmission technology can provide the application relay path for the service transmission in addition to the default path. While expanding the transmission bandwidth, it can effectively avoid the random network congestion and provide reliable transmission for conversational real-time HD video service. However, in the multipath relay transmission, the service QoE does not depend on the transmission quality of a single path but also depends on the transmission control of multiple paths under the coupling relationship. Then, overcoming the challenge of how to use the reasonable load distribution strategy to guide service transmission has become the key of service QoE evaluation. Therefore, in solving the above problems, the main research work and contributions of this paper are as follows:

(1) Without changing the existing network structure, we used P2P and SDN (software defined network) technology for reference [2,3]. Through the deployment of a relay controller and relay server in the backbone network, we used a software defined overlay network to solve the end-to-end transmission bottleneck problem and provided a relay service to meet the needs of users' QoE for conversational real-time HD video service;

(2) According to the types of network service defined by 3GPP (3rd Generation Partnership Project), we analyzed the impact of key QoS indicators on the transmission quality of conversational service. Then, we used the subjective evaluation method of service quality provided by ITU-T (ITU Telecommunication Standardization Sector) for reference [4] and built the mapping relationship between the service QoE and QoS indicators based on expert experience and the user's subjective perception;

(3) Considering the coupling relationship among the sub-paths in the process of multipath relay transmission, we used the theory of fuzzy cooperative games to design the service load distribution strategy. The strategy combines real-time feedback information of each sub-path and uses the reward function to dynamically adjust the load distribution, while the reliable transmission meeting the QoE requirements of service can be obtained under ensuring the service delay.

In the rest of the paper, Section 2 summarizes the related works, and Section 3 introduces the multipath relay transmission scene. The load distribution strategy and the mapping relationship between QoE and QoS of service are presented in Section 4. Our results are shown in the experiments in Section 5. Our discussion and contribution are shown in Sections 6 and 7.

## 2. Related Works

According to the definition of QoE, we analyzed the influencing factors of QoE in video streaming service [5–7]. The influencing factors of the service QoE are divided as shown in Figure 1.

As shown in Figure 1, the influencing factors of QoE evaluation for video streaming service include uncontrollable subjective factors and controllable objective factors. The uncontrollable subjective factors are mainly related to user sentiment, user experience, and user expectation for service [8]. The controllable objective factors mainly involve the end-to-end QoS guarantee mechanism and end-to-end service quality at the technical level as well as the service settings and service billing standards at the non-technical level [9,10]. Combined with the division of influencing factors of video streaming service quality, the video streaming service quality evaluation methods mainly include the subjective evaluation method, objective evaluation method, and pseudo-subjective evaluation method.

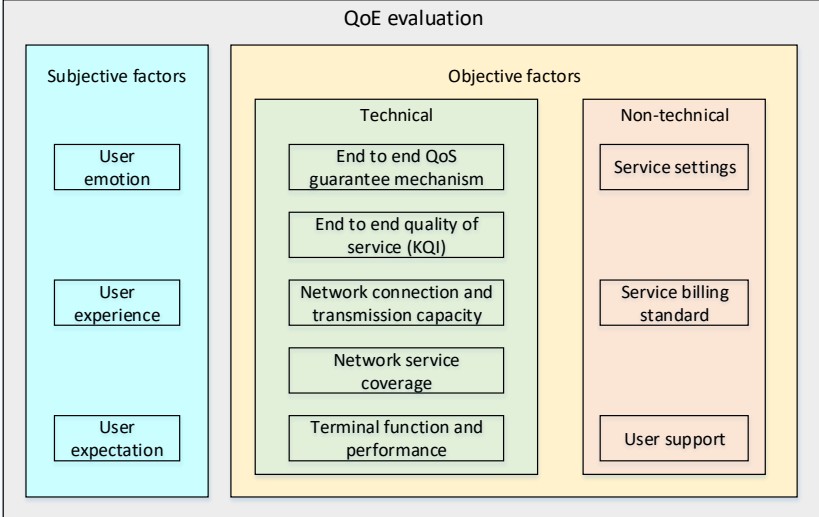

**Figure 1.** The Classification of QoE (Quality of Experience) influencing factors for video streaming service.

The subjective evaluation method is mainly to select suitable evaluators within a specific range of conditions and use their subjective feelings to the test video image to determine the video transmission effect. The radio communications sector of international telecommunication union (ITU-R) proposes two standardized subjective evaluation methods which are SSCQE (Single Stimulus Continuous Quality Evaluation) and DSCQS (Double Stimulus Continuous Quality Scale) [11,12]. Based on the description of the subjective testing process and content, these two methods use the MOS scoring method to subjectively evaluate the test video images and combine the expert experience to analyze the QoS guarantee capabilities of different video streaming services. In addition, some experts and scholars have proposed some improved methods based on these two methods. A method to speed up the evaluation process by adding the feedback information of the evaluator in the subjective evaluation process is proposed in Reference [13]. A method to improve the accuracy of evaluation results by analyzing the effect of video image sequence length on user perception accuracy is proposed in Reference [14]. In general, although the subjective evaluation method can directly and accurately reflect the user's subjective experience of the service, there are disadvantages of cumbersome implementation steps, high test environment requirements, and poor real-time evaluation results.

The objective evaluation method is mainly to compare the uncompressed original video data with the compressed video data to obtain an evaluation index and then use the calculated index value to evaluate the video image quality. At present, the objective evaluation usually adopts two methods of PSNR (Peak Signal Noise Ratio) and PCA (Principal Component Analysis) [15,16]. The PSNR method mainly evaluates the distortion quality of video images, and the PCA mainly evaluates the quality of single-frame images. In addition, there are some quality evaluation methods for specific service types such as video quality evaluation based on the IEEE 802.11wlan multicast service and the WebRTC (Web Real-Time Communication) service [17,18]. In general, although objective evaluation methods are simple and practical, they need to be based on human perception models. It has high requirements on the target accuracy of the algorithm which increases the difficulty of design and implementation.

The pseudo-subjective evaluation method mainly uses artificial intelligence and statistical theory to build complex training models and service evaluation mechanisms for different types of video services through big data processing such as real-time video service quality evaluation based on RNN (Recurrent Neural Network) [19], OTT (Over The Top) video service quality evaluation based on idealized data clustering [20], network video quality evaluation based on fuzzy expert system [21], video service quality evaluation based on S-MDP (Streaming with Markov Decision Processes) [22], and mobile video quality evaluation based on wireless cellular network [23]. In general, although the pseudo-subjective evaluation method combines the advantages of subjective evaluation and objective evaluation to better ensure the real-time and accuracy of video service quality evaluation,

the existing research results focus on the quality of service in the single-path transmission environment. Therefore, our work draws on the ideas of these research results and focuses on the QoE guarantee for conversational real-time HD video service through the adjustment of the load distribution strategy in the multipath relay transmission scenario.

## 3. The Multipath Relay Transmission Scenario

### 3.1. The Software Defined Overlay Network

For the conversational real-time HD video service, the end-to-end single-path transmission may cause the decrease of available bandwidth due to the random background traffic, which easily causes the random congestion of the network. On the one hand, due to the limitation of network node performance and storage space, when the rate of port forwarding data is lower than the arrival rate of data packets, the subsequent arriving data packets will be discarded because the storage space is full, or they will be required to be resent due to the timeout of forwarding which reduces the network efficiency and makes the situation of network congestion more serious. On the other hand, busy network node traffic may increase the time that data packets are cached in the node which may cause large changes in the data arrival rate and network jitter. In this case, if jitter is eliminated, the de-jitter delay may increase which will cause the overall delay to fail to meet the strict real-time constraints of the service, and the QoE of the service will not be guaranteed.

Based on the software defined overlay network, the global network resources can be scheduled and managed from the control plane. On the premise of path generation and controllable selection, the end-to-end transmission bottleneck problem can be solved by adding parallel transmission paths and by using bypass retransmission so as to provide users with a relay service that meets the real-time constraints [24]. The network topology of a software defined overlay network is shown in Figure 2.

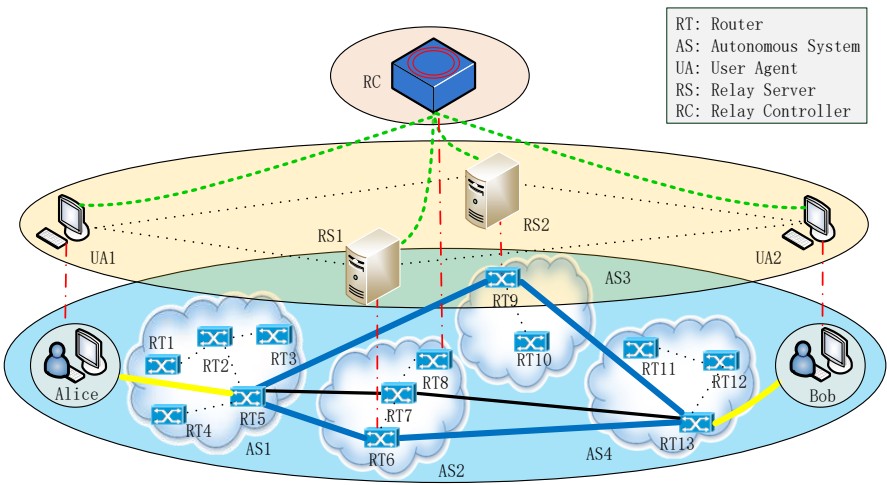

**Figure 2.** The topology of a software defined overlay network.

As shown in Figure 2, the function of software defined overlay network is to build a multipath relay transmission service based on the overlay network without changing the existing network architecture. The network node is the relay node supporting the service media forwarding, and the network link is the end-to-end link of the application layer [25,26]. The network element components in this network structure work in the application layer overlay network of the TCP/IP protocol group, and a logical overlay network is built on the underlying bearer network. These network element components are ordinary network devices with a TCP/IP five-layer protocol stack connected to the bearer network which can be located in data center servers, CDN servers for media storage and distribution, cloud servers, etc. The functions of the main network element component are as follows:

A.    UA (User Agent): It is mainly responsible for the interaction of end users and provides users with an interface to the overlay network architecture. The work content of the UA includes two aspects: One is the processing of control process, mainly including the transmission request of end users, the path information interaction of multipath transmission. The other is the transmission of media data, mainly including the load distribution and sub-stream reorganization;

B.    RS (Relay Server): It is mainly responsible for the aggregation and forwarding of media data among UAs. According to the idea of SDN, all behaviors of RS are under the centralized control of an RC. The work content of an RS mainly includes two aspects: One is the processing of control process which is mainly the interaction of relay transmission path information with the RC. The other is the transmission of media data which is mainly the multipath relay transmission of media data;

C.    RC (Relay Controller): It is mainly responsible for the management and control of all network element components in the overlay network from the whole network. The work content of the RC is divided into two stages: generating the global multipath transmission resource view in the initial stage and completing the scheduling and allocation of transmission resources; and processing the transmission request of a UA in the multipath transmission stage, performing the multipath relay transmission of media data, monitoring the quality of multipath transmission, and dynamically adjusting the transmission strategy according to the transmission feedback.

The transmission scenario is mainly oriented to the media relay service, rather than content distribution services like P2P [27]. We used relay controllers deployed in the backbone network to separate media transmission and control and used relay servers to store and forward media service.

*3.2. The Application Layer Multipath Relay Service*

In the process of multipath relay transmission, the main factor that affects resource aggregation and link anomaly tolerance is the degree of transmission path intersection. For the aggregation of transmission resources, if there is an intersecting link between two sub-paths of parallel transmission, the aggregation of transmission resources will be smaller because of the shared link resources. In addition, if the degree of path intersection is low, the transmission resources of differential links can be fully aggregated. For the link anomaly tolerance, if two sub-paths of parallel transmission are randomly congested in the intersecting link, it will cause two sub-paths to be congested at the same time. If the degree of path intersection is low, it can improve the transmission complementary fault tolerance between different sub-paths and reduce the impact of random congestion on the quality of service. Therefore, in order to reduce the impact of path intersection on the service transmission capacity, we chose the disjoint path to perform the multipath relay transmission of the service according to the current situation where there were many disjoint links between the bearer networks deployed by different operators. In this way, if the service terminal access network was not the transmission bottleneck, when the UA established a real-time session, the transmission path could select and use the application relay path in addition to the default route provided by the network layer. The relay path expansion of the overlay network is shown in Figure 3.

As shown in Figure 3, when A (Alice) and B (Bob) establish a real-time session, if RT1 and RT3 of the terminal access network link are not the bottleneck of transmission, the multipath relay service can select disjoint path1 and path2 to perform parallel transmission of media data which can effectively improve the transmission capacity of the service. For example, if the real-time transmission bandwidth of path1 (Default path) is having difficulty meeting the service QoE requirements due to the random congestion of the network, because path2 expands the transmission bandwidth, the service QoE can still be guaranteed under the control of the appropriate load distribution strategy.

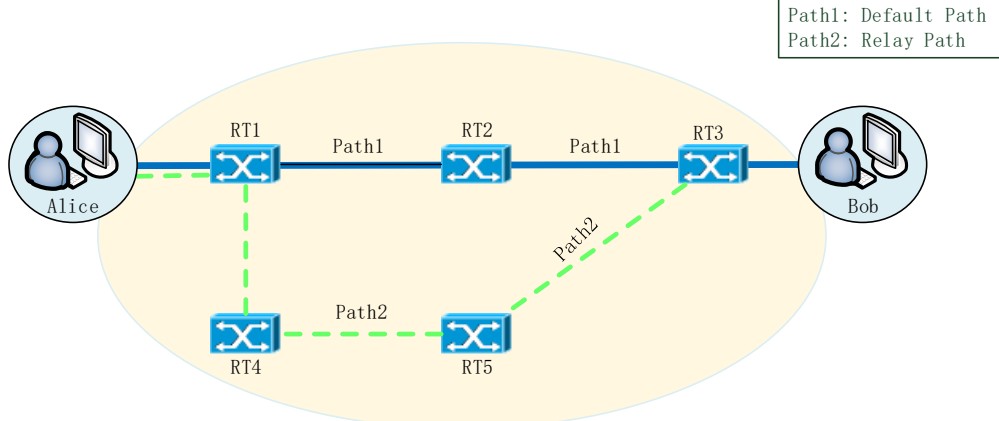

**Figure 3.** The topology of the application layer multipath relay service.

## 4. The Service QoE Evaluation

### 4.1. The Mapping Relationship between Service QoE and QoS

At present, 3GPP divides network services into four types: conversational class, streaming class, interactive class, and background class [28]. Based on the description of conversational service by 3GPP, we obtained the QoE characteristics of the service according to the transmission requirements. The QoE characteristics of conversational service are shown in Table 1.

**Table 1.** The QoE (Quality of Experience) characteristics of conversational service.

| Conversational Service | QoE Characteristics |
| --- | --- |
| The service transmission characteristics | Small end-to-end delay<br>Upstream and downstream traffic are symmetrical or almost symmetrical |
| The main QoS (Quality of Service) indicators of service | Delay (strictly limited)<br>Jitter (strictly limited)<br>Packet loss (restriction) |
| The service priority | Highest priority<br>Guaranteed delay<br>Reserved bandwidth |

As shown in Table 1, conversational service is a two-way video streaming service with strict real-time constraints (The typical service is internet phone). On the premise of meeting the requirements for service bandwidth, the main QoS indexes for evaluating service quality include delay, jitter, and packet loss:

A.　Delay: Based on the diversity of network communication structure and underlying transmission protocol, the delay of conversational service is composed of fixed delay and variable delay. The fixed delay mainly includes coding and packing delay; it depends on the performance of terminal equipment, the compression algorithm, and the service data volume. The variable delay mainly includes the bearer network transmission, node queuing, service processing, and jitter removal delay; it depends on the port rate of the device, the load capacity of the network, the status of the transmission link, and the QoE mechanism. When there is random congestion in the network, the high delay will have a serious impact on the picture quality and fluency of the service; we took the delay of transmission path as a key index of the service quality evaluation without considering the terminal performance and codec mechanism;

B.　Jitter: The jitter index is closely related to the delay index. For packet transmission, the time-varying path state may cause the packet to arrive at the receiver cache inconsistent of

time. When the receiver removes the impact of jitter on the quality of service, it may cause the network random congestion due to the packet loss retransmission. Therefore, we took the jitter as an important index of the service quality evaluation;

C.　Packet loss: During conversational service transmission, the main cause of packet loss is the random network congestion caused by insufficient service bandwidth. When there is random congestion in the network, the increase of queuing delay will increase the probability of packet loss due to the limitation of the buffer capacity which may lead to a short pause or mosaic for the service. Because packet loss is inevitable in the bearer network, we took the packet loss as a restrictive index of service quality evaluation under the premise of not exceeding the user tolerance.

According to the introduction of ITU-T.TP 800 for MOS evaluation method, we divided the subjective evaluation of conversational real-time HD video service into five levels. The specific division standards are shown in Table 2.

**Table 2.** The MOS (Mean Opinion Score) evaluation standard.

| MOS Grade | QoE Evaluation | Degree of Damage |
| :---: | :---: | :---: |
| 5 | Excellent | No perceiving |
| 4 | Good | Can be perceived but not serious |
| 3 | Common | Slight |
| 2 | Bad | Serious |
| 1 | Very bad | Very serious |

As shown in Table 2, according to the classification of MOS grade, we used the currently widely used DSCQS method for subjective testing. First, we chose 30 users in the age range of 20–30 as testers (the identities of the users were all students in school, they usually choose real-time video service actively), and helped them master the basic test skills through relevant test training. Second, we took a randomly recorded HD video session of 100 s as the test sample. After the video preprocessing, we used 10 s as the playback time and 2 s as the playback interval. The tester subjectively scored the video clips during the playback interval. Then, in order to ensure the reliability of the test results, we performed cyclic playback and test scoring on the video clips under different network performance and obtained a total of 15,000 reliable evaluation data. Finally, based on the expert experience introduced in ITU-T BT.500-13, we averaged the MOS scores of a single video clip. The subjective test results are shown in Figure 4.

As shown in Figure 4, according to the subjective test results, we objectively and accurately obtained the user's subjective perception by establishing the mapping relationship between the service QoE and QoS:

A.　As shown in Figure 4a, for the user subjective perception, a delay ≤150 ms can obtain a good quality of service experience, and the smaller the delay, the better the subjective feeling of users. However, when a delay is ≤80 ms, the users can hardly perceive the impact of the delay on service quality. Even if the delay is further reduced, the users' subjective feelings did not change significantly;

B.　As shown in Figure 4b, when the jitter ≥60 ms, it was easy to cause a large delay in the process of eliminating jitter which will cause a delay to exceed the upper limit that users will tolerate. When the jitter was ≤40 ms, the service quality was able to meet the needs of users combined with the close relationship between delay and jitter;

C.　As shown in Figure 4c, when the packet loss ≥6%, the poor service quality was intolerable to users. When 3% ≤ packet loss ≤ 6%, although there were some slight voice pauses or unclear video images during the conversation, the users could still get a satisfactory quality experience. Especially when packet loss ≤3%, the impact of packet loss on the service quality can be ignored.

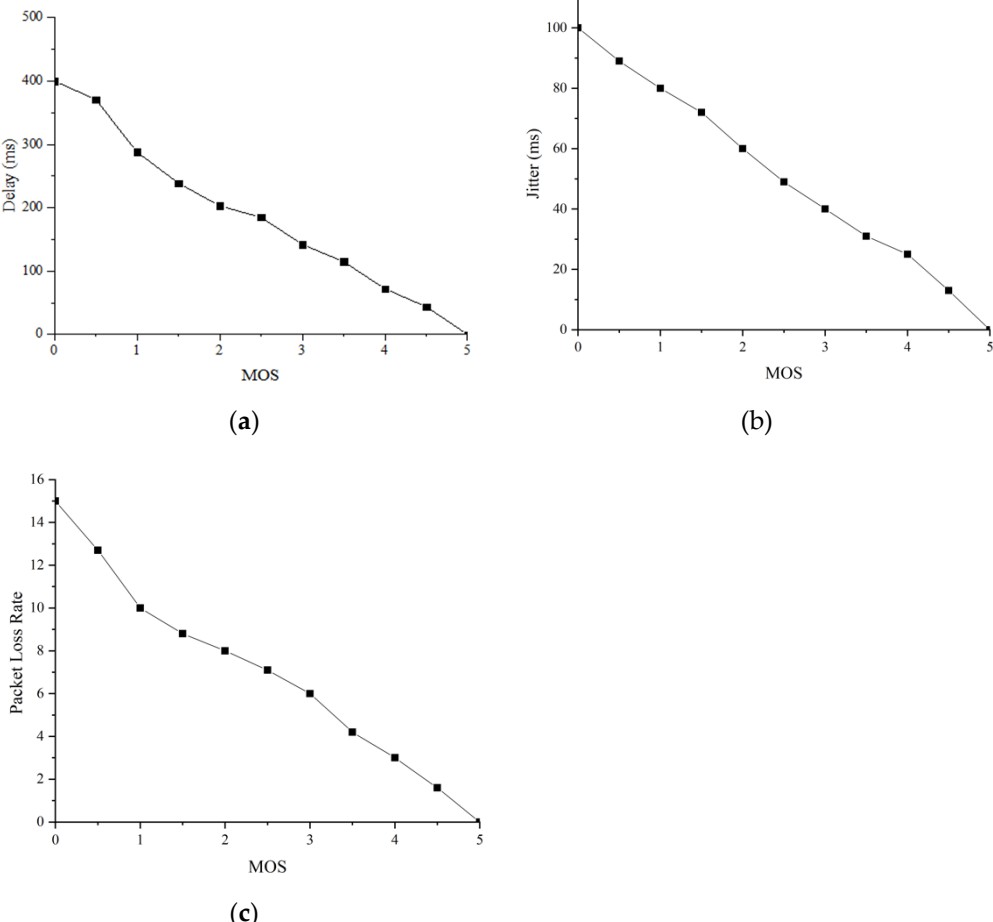

**Figure 4.** The mapping relationship between the service MOS and delay, jitter, and packet loss. (**a**) is the test result of MOS and delay. (**b**) is the test result of MOS and jitter. (**c**) is the test result of MOS and packet loss.

### 4.2. The Service Load Distribution Strategy

In the scenario of multipath relay transmission, when we discover and choose to allocate multiple transmission links for conversational real-time HD video service, each sub-path has a resource competition relationship in the process of parallel transmission which makes the transmission strategy selected by each sub-path affect the overall transmission quality of the service while determining its own transmission effect. Therefore, we used fuzzy cooperative game theory for reference and used the established six tuple mathematical model to describe the transmission control process of the service based on the coupling relationship among the sub-paths in the multipath relay transmission:

$$(X, Y, Y^*, Z, f, F) = \begin{cases} X = \{x_1, x_2, \ldots, x_n\} \\ Y = \{y_1, y_2, \ldots, y_n\} \\ Y^* = \{y_1^*, y_2^*, \ldots, y_n^*\} \\ Z = \{z_1, z_2, \ldots, z_n\} \\ f = f(z_1, z_2, \ldots, z_n) \\ F = \{f_1, f_2, \ldots, f_n\} \end{cases} \tag{1}$$

As shown in Formula (1), $X$ is a set of multiple optional sub-paths for the multipath relay transmission; each sub-path can obtain link status information through real-time measurement such as delay, jitter, packet loss, etc; $Y$ is the set of transmission strategies that each sub-path chooses arbitrarily according to its own link state, $y_i^*$ is the set of all feasible strategies for sub-path $x_i$; $Y^*$ is the restrictive

setting of the transmission strategy according to the actual transmission situation of the service; $y_i^*$ is the set of restrictive strategies of sub-path $x_i, y_i^* \in y_i (i = 1, 2, \ldots, n)$; $Z$ is the set of transmission policies actually adopted by each sub-path which can be selected from $Y^*$; $f$ is the overall objective function of the system which represents the transmission quality of service according to the load distribution strategy selected by each sub-path; and $F$ is the set of objective functions of each sub-path, where $f_i = f_i(z_1, z_2, \ldots, z_n)$ is the objective function of sub-path $x_i$.

Since the optimal transmission quality of each sub-path is closely related to the load distribution strategy adopted by itself and other sub-paths, when the random background traffic changes the QoS of the sub-path, we dynamically adjusted the load distribution of each sub-path by setting the reward function on the basis of the initial transmission strategy so as to better guarantee the transmission quality of the service. During the multipath relay transmission process, the RC is responsible for delivering the load distribution strategy to the relevant UA and RS. The RC uses global multipath transmission resource management and real-time path transmission quality monitoring to generate the strategy based on the number of paths and key QoS indicators acquired as input conditions. According to the coupling relationship between multipath transmission paths, the objective function of the system is to ensure that the service delay meets strict real-time constraints while reducing packet loss, and the objective function of each sub-path obtains more transmission resources based on its own transmission capacity. In this case, according to the dynamic network performance changes, in order to avoid the increase in packet loss caused by the slow path transmission timeout, a single-path delay $\leq 150$ ms is used as the limiting condition for parallel transmission, and $L_{p_i}^{min}$ is used as the constraint for adjusting the multipath load distribution strategy. The design of the reward function is as follows:

(1) When the transmission strategy chosen by each sub-path is not the best one, it will cause the loss of service quality. We used $\theta_i = \theta_i(z_1, z_2, \ldots, z_n)(i = 1, 2, \ldots, n)$ to express the responsibility of each sub-path for the loss of service quality so as to obtain the relationship between the optimal overall objective function and each sub-path's objective function:

$$f_i' = f_i'(z_1, z_2, \ldots, z_n) = F_i' - \theta_i(F' - f)(i = 1, 2, \ldots, n) \tag{2}$$

As shown in Formula (2), $f_i'$ is the objective function of each sub-path; $(z_1, z_2, \ldots, z_n)$ is the actual transmission strategy adopted by each sub-path; $F'$ is the maximum value obtained by the overall objective function $f$, when each sub-path executes the optimal strategy $(z_1', z_2', \ldots, z_n')$; $F_i'$ is the maximum value of the objective function of each sub-path, when the $F'$ maximum value is satisfied. If $(z_1, z_2, \ldots, z_n) \neq (z_1', z_2', \ldots, z_n')$, because each sub-path does not adopt the optimal strategy, the overall objective function obtained will be lower than the optimal value with a great probability:

$$\begin{aligned} \theta_i &= \frac{f_i'(z_1', z_2', \ldots, z_n') - f_i'(z_1, z_2, \ldots, z_n)}{f(z_1', z_2', \ldots, z_n') - f(z_1, z_2, \ldots, z_n)} \\ &= \frac{F_i' - f_i'}{F' - f}(i = 1, 2, \ldots, n) \end{aligned} \tag{3}$$

(2) When the initial load distribution strategy adopted by each sub-path is not the optimal strategy, we used $a_i$ to adjust the load distribution of each sub-path, so as to ensure that the overall transmission quality of the service still satisfied $F'$. Formula (2) can be changed as follows:

$$f_i' = f_i'(z_1, z_2, \ldots, z_n) = F_i^0 + a_i(F' - F^0) - \theta_i(F' - f) \tag{4}$$

(3) From the above analysis, when $(z_1, z_2, \ldots, z_n) \neq (z_1', z_2', \ldots, z_n')$, we can get $F^0 < F'$. According to this conclusion, we set-up a set of reward functions $K_i = K_i(z_1, z_2, \ldots, z_n)(i = 1, 2, \ldots, n)$ and obtained a new objective function by the superposition of the initial function of each-sub path:

$$f_i' = f_i'(z_1, z_2, \ldots, z_n) = f_i^0 + K_i(i = 1, 2, \ldots, n) \tag{5}$$

As shown in Formula (5), we adjusted the initial objective function $f_i^0$ of each-sub path through the reward function $K_i$, so that we can make the actual strategy $(z_1, z_2, \ldots, z_n)$ of each-sub path closer to the theoretical optimal strategy $(z_1', z_2', \ldots, z_n')$ according to the QoS conditions of the path in real time, and then ensured that the overall transmission quality of the service satisfied $F'$.

## 5. Experiment

### 5.1. The Simulation Environment

We used OMNeT ++ (Objective Modular Network Testbed in C ++) to build simulation scenarios. The simulation process was based on a multipath relay service. By loading the INET (Network transport control) packets, the relay servers, relay controllers, and service terminals were configured under the "network" and "application" files in "omnet/inet/resource" [29]. The topology of the test network is shown in Figure 5.

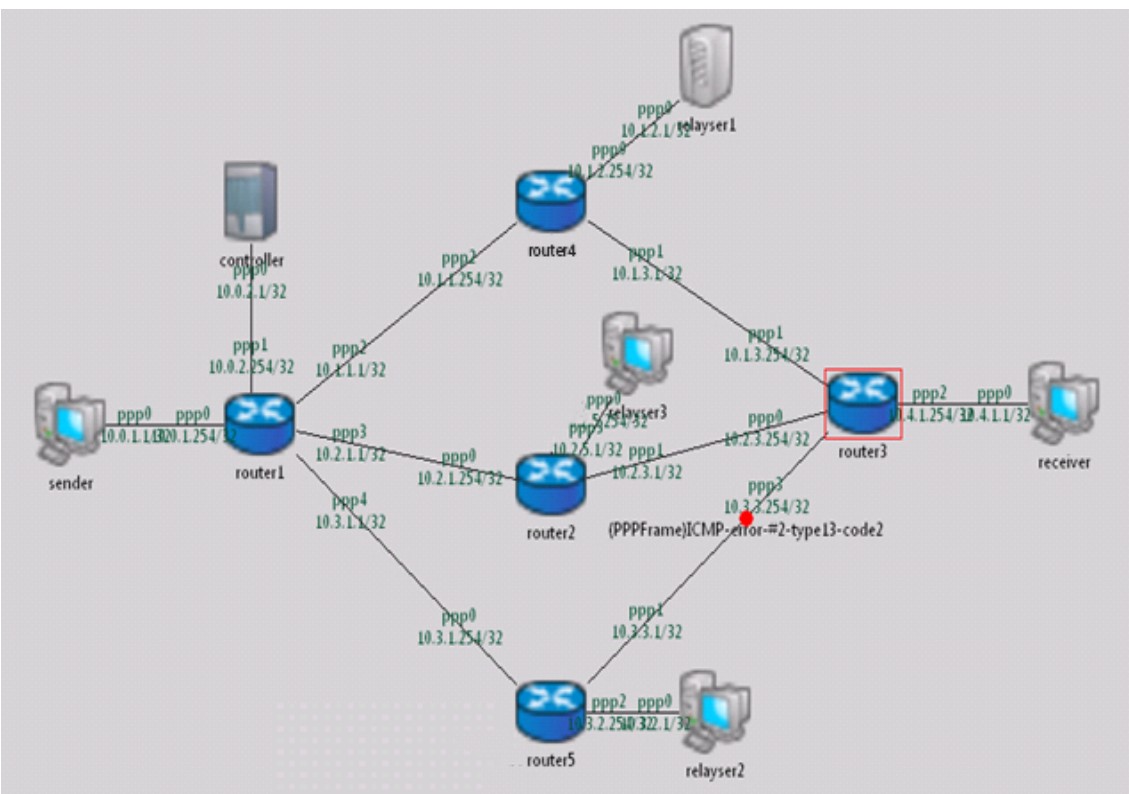

**Figure 5.** The topology of the test network.

As shown in Figure 5, sender and receiver are service terminals based on the UA configuration, and router 1–5 are relay routes that perform service data transmission at the network layer. After the service terminal initiates a real-time HD video service request, the RC load deployed on router1 processes the service terminal's multipath transmission request. In the initial stage, after the generation and management of global multipath transmission resources, RC uses the RS deployed on router 2, router 4, and router 5 to allocate three relay transmission paths for service terminals:

A.      Path 1 (The default path): Sender-> router1-> router4-> router3-> receiver;
B.      Path 2: Sender-> router1-> router2-> router3-> receiver;
C.      Path 3: Sender-> router1-> router5-> router3-> receiver.

## 5.2. The Sample Test

In order to simulate a real network environment, we added random background traffic in the process of link initialization configuration according to the established simulation scenario and monitored the initial link state when the service session was established. The QoS information of the three sub-paths is shown in Table 3.

**Table 3.** The QoS information of the three sub-paths.

| Path | Delay | Jitter | Packet Loss Rate | Bandwidth |
|------|-------|--------|------------------|-----------|
| path1 | 103 ms | 24 ms | 2% | 10 Mbit/s |
| path2 | 121 ms | 20 ms | 4% | 13 Mbit/s |
| path3 | 89 ms | 16 ms | 1% | 12 Mbit/s |

As shown in Table 3, according to the sub-path QoS information obtained by monitoring, we used QoE as the evaluation standard to establish the judgment matrix of the QoS index by comparing the impact of different QoS indexes on the service transmission quality [30,31]:

$$Q = \begin{bmatrix} 1 & 1/3 & 1/5 & 2 \\ 3 & 1 & 1/4 & 3 \\ 5 & 4 & 1 & 6 \\ 1/2 & 1/3 & 1/6 & 1 \end{bmatrix} \tag{6}$$

After the normalization of Formula (6), we obtained the weights of different QoS indexes by solving the eigenvectors of the judgment matrix as follows:

$$Q = \begin{bmatrix} 0.074 & 0.044 & 0.095 & 0.118 \\ 0.222 & 0.130 & 0.118 & 0.176 \\ 0.371 & 0.521 & 0.472 & 0.353 \\ 0.037 & 0.044 & 0.079 & 0.059 \end{bmatrix} \tag{7}$$
$$W = (0.444, 0.078, 0.154, 0.054)^T$$

As shown in Formula (7), we referred to the general research methods adopted in recent years in the research work of service QoE evaluation [32] and obtained the QoS values of three sub-paths using the linear weighting method after normalization:

$$\begin{aligned} P_{QoS} &= W_D \times D + W_J \times J + W_L \times L + W_B \times B \\ p_1^0 &= 0.444 \times 0.103 + 0.078 \times 0.024 + 0.154 \times 0.02 + 0.054 \times 10 \approx 0.591 \\ p_2^0 &= 0.444 \times 0.121 + 0.078 \times 0.020 + 0.154 \times 0.04 + 0.054 \times 13 \approx 0.764 \\ p_3^0 &= 0.444 \times 0.089 + 0.078 \times 0.016 + 0.154 \times 0.01 + 0.054 \times 12 \approx 0.691 \end{aligned} \tag{8}$$

As shown in Formula (8), according to the obtained QoS scores of each sub-path, we get the initial load distribution ratio of the three sub-paths after quantitative processing as follows:

$$\begin{aligned} p_1^0 &\approx 28.9\% \\ p_2^0 &\approx 37.3\% \\ p_3^0 &\approx 33.8\% \end{aligned} \tag{9}$$

As shown in Formula (9), after judging that the sub-path satisfies $D_{p_i} \leq 150\ ms$, we used $L_{p_i}^{min}$ as the constraint condition in combination with packet loss of each sub-path. Based on Formulas (4) and (5), we get the three sub-path adjusted load distribution as follows:

$$p'_1 = \frac{p_1^0 \times L_{p3}^{min}}{L_{p_1}} \approx 14.5\%$$

$$p'_2 = \frac{p_2^0 \times L_{p3}^{min}}{L_{p_2}} \approx 9.3\% \tag{10}$$

$$p'_3 = p_3^0 + (p_1^0 - p'_1) + (p_2^0 - p'_2) \approx 76.2\%$$

As shown in Formula (10), we implement the bypass retransmission on the sub-path with low delay according to the adjusted load distribution strategy, and use $\sum\limits_{i=1}^{3} p'_1 \times L_{p_i} = L$ to check the packet loss of the service, so as to ensure that the real-time packet loss $\leq 3\%$.

In order to verify the application effect of the load distribution strategy designed in this paper, we randomly selected online live video from the open source project COIN (Comprehensive Instruction Video Analysis) as the sample for the simulation test without considering the terminal processing capacity [33]. In the real-time HD video session of 150 s, we used the designed load distribution strategy to perform service transmission in the simulation scenario. By monitoring the transmission process, we obtained the test results as shown in Figure 6.

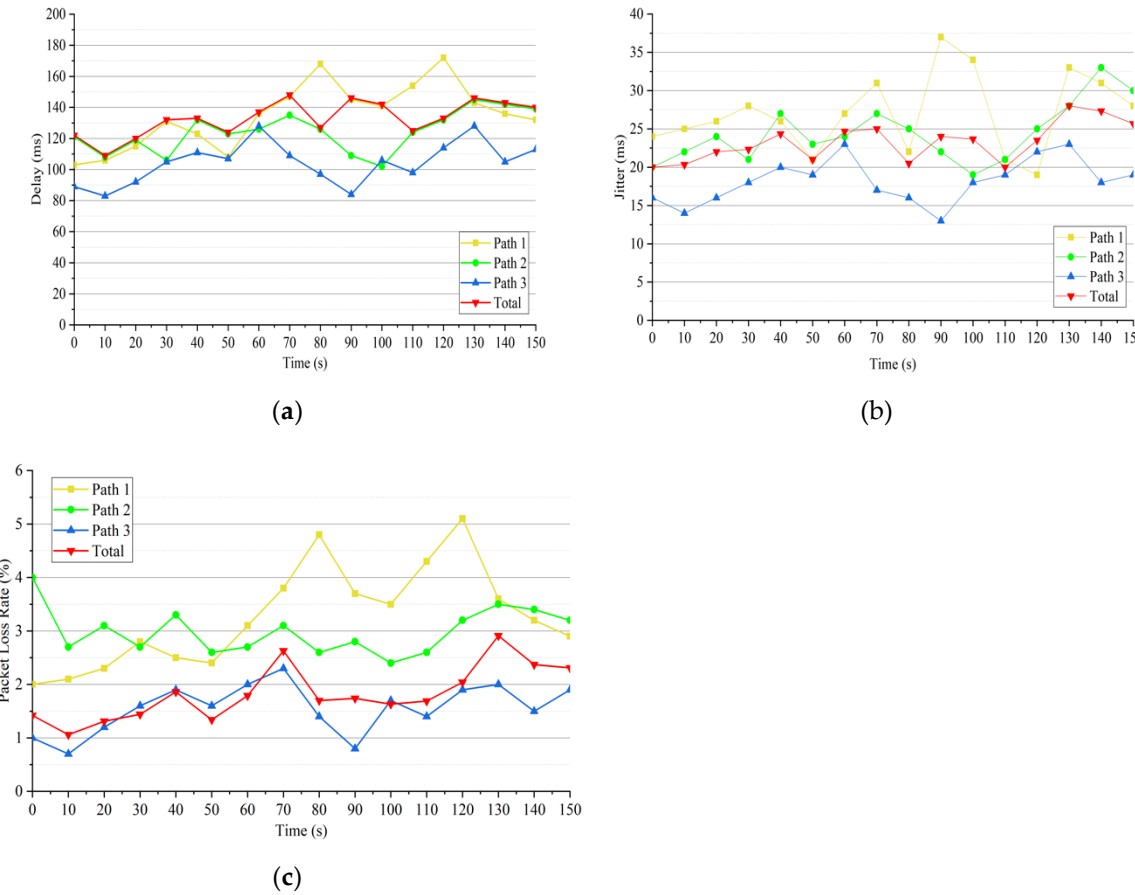

**Figure 6.** The test sample QoS information monitoring. (**a**) is the real-time monitoring delay of the three sub-paths and the overall service delay obtained after the adjustment of the load distribution strategy. (**b**) is the real-time monitoring jitter of the three sub-paths and the overall service jitter obtained after the adjustment of the load distribution. (**c**) is the real-time monitoring packet loss of the three sub-paths and the overall service packet loss obtained after the adjustment of the load distribution.

As shown in Figure 6, affected by random background traffic, the path status of the three sub-paths was in a state of constant change. During the test lasting for 150 s, we can see that path1 (the default path) had a high delay in different time periods, while paths 2 and 3 (the assigned relay paths) always had a low level of delay. In order to guarantee the real-time transmission quality of the service, we adopted two methods to adjust the load distribution according to the measurement results of the real-time path information:

A.  When the service session starts, through real-time monitoring of the path information, we used $L_{p_i}^{min}$ as a constraint under the condition of judging the three sub-paths' delay of 150 ms and, according to Formulas (8)–(10), to obtain instant load distribution strategy;

B.  When monitoring the path1 of 150 ms in 75 s, where its jitter changes significantly and the packet loss shows an increasing trend, we determined that path1 may have network congestion based on the mapping relationship between service QoE and QoS. In this case, since the data packets sent in sequence in multipath transmission will be used the ACK returned by the slow path as an acknowledgment response, in order to ensure the strict real-time constraints of the service, we set the load distribution ratio of path1 as 0 based on the limiting condition designed in the reward function. Then, we enabled the bypass retransmission mechanism and used Formula (9) and (10) to adjust the transmission strategy of other sub-paths with a delay of 150 ms according to the constraint designed in the reward function.

Based on the above two methods, we can see from the test results that when random congestion occurred on a single path during the test, the load distribution strategy we designed can still guarantee the transmission quality of the service. In addition, in order to ensure the accuracy and objectivity of the test results, we randomly selected 200 online live videos from the COIN as test sample sequences and reconfigured the simulation environment. Through the real-time monitoring of the network status, the test was conducted in two states of the network: idle and busy. Based on the similarity of the study's objective and purpose, under the same test environment, we conducted a comparative test with the single-path transmission method adopted in Reference [19]. The comparison test results are shown in Figure 7.

As shown in Figure 7, through comparison tests, we can see that when the network was idle (as shown in Figure 7a), the multipath relay transmission was affected by the time complexity of the algorithm compared to the single-path transmission. Although the delay was slightly higher, it still met the strict real-time constraints of the service, and through the adjustment of the load distribution strategy, it reduced jitter and reduced packet loss which effectively guaranteed the stability of transmission. When the network was busy (as shown in Figure 7b), the network was congested due to the addition of random background traffic, and the transmission quality of the single-path transmission was difficult to be guaranteed under the premise of limited bandwidth resources. The multipath relay transmission still effectively guaranteed that the key QoS indicators met the service transmission quality requirements through the adjustment of the load distribution strategy.

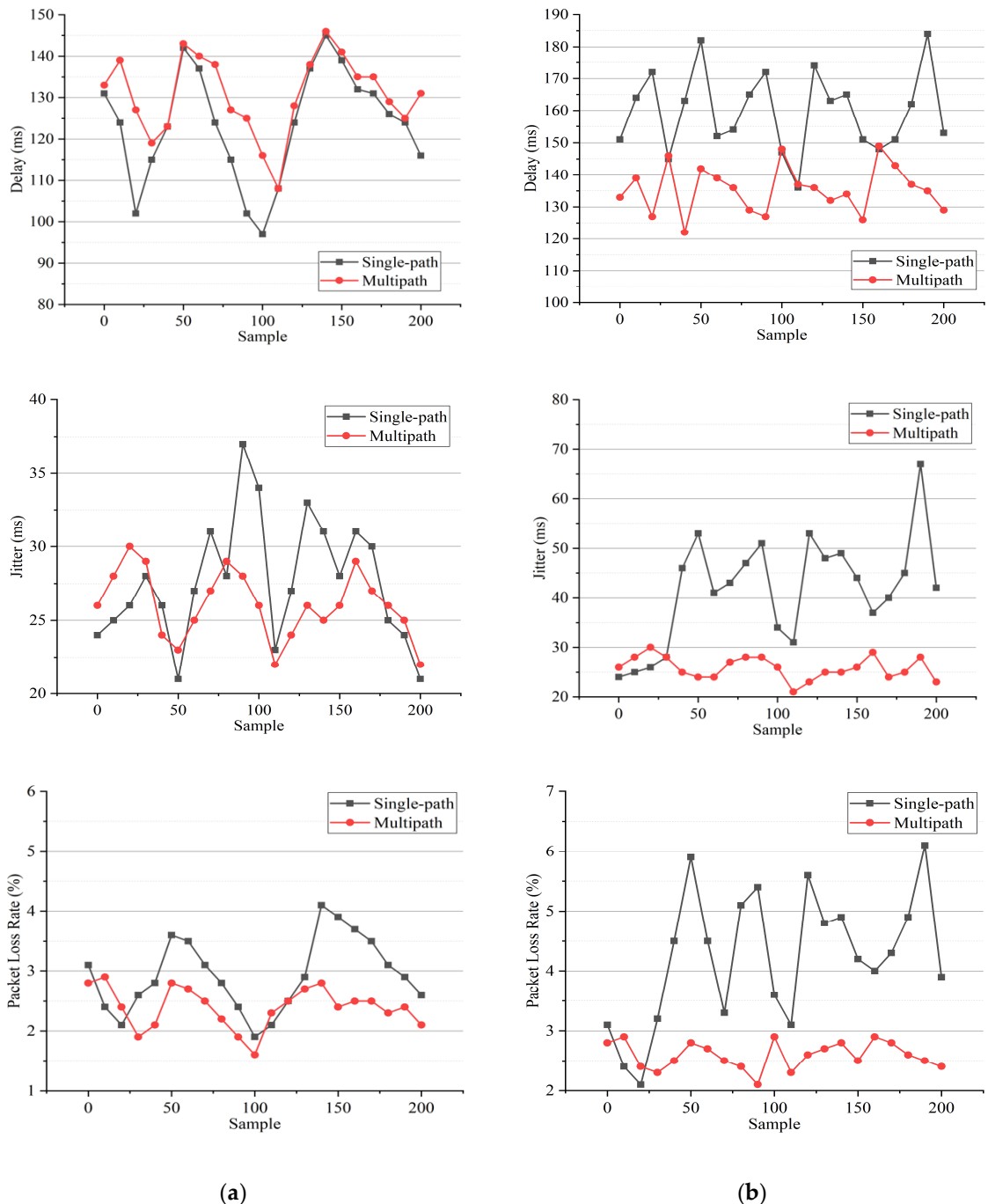

**Figure 7.** Comparison of sample test results between the single-path transmission and multipath relay transmission. (**a**) is the comparison result of the two transmission methods in the network idle state. (**b**) is the comparison result of the two transmission methods in the network busy state.

## 6. Discussion

For the QoE evaluation of conversational real-time HD video service, our purpose was to build an application layer multipath relay service to avoid the impact of random network congestion on service quality, and we used a designed load distribution strategy to ensure that the transmission quality of the service met users' needs. However, due to the lack of relevant references and technical documents at this stage, there are still some defects in our research work:

(1) Providing an application layer relay service for conversational real-time HD video service involves many problems such as network resource scheduling, relay path generation, and relay path maintenance. The solutions to these problems have been detailed in our team's other published papers [25,26]. Therefore, in the research work of this paper, we used path generation and measurement information as known input conditions to adjust the real-time transmission load distribution strategy. At the same time, considering the performance differences of relay nodes, we evaluated the transmission quality of a single relay path as a whole;

(2) According to the concept of QoE, the QoE evaluation of service is the description of users' perception of service quality. Because of the uncontrollability of users' subjective perception, the current mainstream QoE evaluation methods of video streaming media service mainly adopt the way of big data processing. In order to achieve the goal of service QoE quantification, we established the mapping relationship between service QoE and QoS based on expert experience and users' subjective perceptions after analyzing the QoE characteristics of conversational service. However, due to the limitation of experimental conditions, there may be some errors between our quantitative results and reality;

(3) When using multipath relay transmission technology to carry out the load distribution of service data, the packet disorder caused by the difference between paths is inevitable, which may result in the performance of multipath transmission lower than that of single-path transmission. From the simulation test results, we can see that the QoS index of single-path transmission is not different from that of multipath transmission when the cache condition is not considered and the network is idle, while the ability of multipath transmission to deal with the random network congestion is more obvious when the network is busy. Therefore, in order to determine which network state should use the multipath relay transmission for service load distribution, we need to set the critical condition that the multipath transmission is better than the single-path transmission by comparing the service transmission rate;

(4) The control plane is the core of the software defined overlay network management. When the control plane needs to expand the scale of the network, the control plane control load will change accordingly, then the control plane may also become a performance bottleneck and affect the quality of service transmission. Since scalability and time complexity are important factors to measure the performance of a method, the performance of the method we design mainly depends on the network scale of the control plane. Limited by the test conditions, the performance of the method may be affected in the application of actual complex networks.

## 7. Conclusions

Based on the fact that conversational service has not matured and open QoE transmission technology, our contribution was the proposal of a load distribution strategy with QoE transmission guarantee for conversational real-time HD video service. The main content of this strategy was to use multipath relay transmission technology to solve the impact of random network congestion on the quality of service. First of all, we analyzed the transmission characteristics of service QoE and completed the quantification of service QoE by establishing the mapping relationship between service QoE and QoS. Second, combining with the coupling relationship between paths in multipath relay transmission, we used the fuzzy cooperative game theory to describe the service transmission process and calculated the QoS index weight and each sub-path weight according to the real-time path QoS information so as to obtain the initial load distribution strategy of the service. Finally, considering that the random background traffic may change the network's performance, we used the set reward function to dynamically adjust the load distribution strategy based on the time-varying path state. The simulation results show that compared with conventional single-path transmission, this strategy takes advantage of application layer multipath relay service. On the basis of meeting the service bandwidth requirements and data transmission rate, it can give full play to the transmission capacity

of each sub-path according to the real-time network state so as to significantly improve the service QoE with low delay.

**Author Contributions:** Conceptualization, Y.Z., W.L., Y.G., and H.L.; Data Curation, Y.Z.; Formal Analysis, Y.Z.; Investigation, Y.Z., Y.G., H.L.; Methodology, Y.Z., W.L., and H.L.; Supervision, W.L.; Validation, Y.Z.; Writing—Original Draft Preparation, Y.Z.; Writing—Review and Editing, Y.Z., W.L., and Y.G. All authors have read and agreed to the published version of the manuscript.

**Funding:** This work was supported by the National Key Research and Development Program of China Grant No. 2018YFB1702000, the National Natural Science Foundation of China Grant No. 61671141, and the Liaoning Provincial Natural Science Foundation of China Grant No. 20180551007.

**Conflicts of Interest:** The authors declare no conflict of interest.

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
