# Peer review of "Research on the Load Distribution Strategy to Meet the QoE Requirements for Conversational Real-Time HD Video Service"

_electronics, doi:10.3390/electronics9050790_

Round 1

Reviewer 1 Report

The article deals with a very interesting topic related to traffic balancing in computer networks. However, I have doubts about the quality of presentation of the content presented in the article and about the structure of the article itself. Below are my comments:

1) The paper requires an extensive editing of English language and style.

2) I have not found information for what type of network the presented solution is dedicated for: campus networks, core networks, multi-operator networks, cellular networks, cable networks, cellular etc.?

3) The authors assume the use of SDN technology - which controller will be used to implement this architecture and which version of the OpenFlow protocol? Not all routers support OpenFlow - have the authors considered such a case in their research?

4) I have the impression that the article focuses on the method of choosing an alternative data transmission path. As I understand the basic path is determined based on routing. So how is the alternative transmission route implemented in the network? How does this affect the routing table? Have the authors considered mechanisms such as ECPM in OSPF.

5) In my opinion, the assumption from line 416 (1) significantly simplifies the simulation environment, in particular the wording from line 423 is incomprehensible: "we set the hop number of relay route as 1 in the simulation test, and evaluate the transmission quality of a single relay path as a whole by ignoring the QoS differences in each section of the relay path; "

6) Can the proposed solution create a routing loop? In my opinion, in the proposed solution, routing loops will appear in case of classic routing, hybrid SDN implementation (available on Extreme Networks devices, for example) or full SDN implementation.

7) On what basis did the authors develop the charts from Figure 4. What does the phrase "the subjective evaluation method of service" from line 246 mean? How did the experts evaluates the quality? How large was the group of end-users who participated in the experiment?

8) The literature review mainly concerns evaluation methods for QoE evaluation of the service. It may be worth rebuilding the article to cover only this aspect.

9) The article does not contain a comparison of the proposed solution with other similar solutions

There are many general formulations and assumptions that simplify the problem. I have cited only a few examples above.

The work contains editing errors, and the whole work should be corrected in this aspect. Here are some examples:

  • Figure 5 is unreadable
  • Figure 1 uses the wording "End to end quality of service (KQI). Should the abbreviation KQI be understood as Key Quality Indicators?
  • The abbreviations used are not explained. For example, what does P2P (line 52) mean - Peer-to-peer, person-to-person?
  • I do not understand the sentence (line 445): "Based on the conversational service has not mature and open QoE transmission technology, our contribution is to propose the load distribution strategy with QoE transmission guarantee for conversational real-time HD video service"

In summary, the article is interesting, but requires major corrections before publishing. You may need to change the title, limit the scope of the topic and rewrite the article.

Author Response

Thank you very much for your work,please see the attachment.

Reviewer 2 Report

Paper organization and presentation issues:

The language of the submission requires a major revision by the authors. There are typos and many grammatical mistakes to the point that some of the sentences are not understandable. Another concern is the presentation of the paper. The actual contribution of the paper starts very late (about page 5 or 6). The explanation in the related work section, for instance, is quite long and disconnected. Excessive use of itemizing takes from the readability of the paper. 

Technical concerns:

How did the authors collect the result for figure 4? There is neither reference another indicator as to whether the authors collected the users' subjective opinion or it is from another paper. In general, subsection 4.1, similar to section 3, does not show any novelty.

The main concern I have is why the authors didn't try multi-path TCP (MPTCP)? There is a large body of the literature using MPTCP for higher throughput communication. Isn't it the goal of this paper? Also, the authors mentioned they are trying to solve the problem without any infrastructure modification. However, they introduced an overlay with a relay controller and relay servers. 

It seems the authors didn't consider the network dynamicity in their evaluation. There is no algorithm to show how does the proposed scheme work and adopt the network's radical behavior shift.

The evaluation is not realistic with lots of configuration being fixed such as the number of paths and the length of paths. The authors have mentioned part of this concern in the discussion. However, showing the results of unrealistic scenarios is misleading.

Author Response

Thank you very much for your work, please see the attachment.

Reviewer 3 Report

This work proposes a load distribution strategy of the multipath relay transmission tasks for conversational real-time HD video service, which within three stages.

However, I think there is room for the authors to enhance how the results are presented in regard to the related work. Section 2 presents extensively the current work in which several related works were discussed  (around 10 references from reference 14 to references 23). Surprisingly, as we continue to read, we expected at least the authors to discuss one or several related works in their experiment discussion, which was not done. The packets ratio, delays and other metrics that were discussed are very good but at least a separate paragraph (table or figure,,..) should try to discuss the proposed approach inline with section 2.

Author Response

(The authors gave the same response as above.)

Reviewer 4 Report

The main weakness of the paper is that the authors did not evaluate the performance of their method when the available bandwidth at each link is changed. Usually, the main reason of performance degradation with VoIP by network congestion is that the available bandwidth at each link changes with bursty background traffic. The authors should add simulation results with the available bandwidth at each link changed substantially.

The second weakness is that the problem is not well described. Basically, the proposed method is to optimize the routes in order to maximize the performance. However, this problem is not well described. The authors should describe their optimization problem with respect to its inputs, outputs, constraints and objective function.

The authors should also add explanation on how the optimization is performed. In order to search for a solution of such a problem, the system needs to collect information from the entire network, solve the problem, and then distribute the results to the entire network. This is not a simple task, and it is usually not easy to measure the link status such as delay, jitter and packet loss. The performance of this optimization may vary according to which node the solution is searched.

The authors should explain how scalable their method is. It is preferable that the time complexity of the algorithm is shown.

The authors should explain what happens if multiple instances of the proposed method are used within the network. They would interfere with each other.

The authors should explain what happens if the traffic in the control plane is delayed due to network congestion.

Author Response

(The authors gave the same response as above.)

Round 2

Reviewer 1 Report

In my opinion, the article can be published in its current form.

Reviewer 2 Report

The authors answered the reviewer's concerns.

Reviewer 3 Report

The comments were addressed accordingly, minor speel check are required

Reviewer 4 Report

Comments are adequately addressed.